# Dietary Factors in the Control of Gut Homeostasis, Intestinal Stem Cells, and Colorectal Cancer

**DOI:** 10.3390/nu11122936

**Published:** 2019-12-03

**Authors:** Federica Francescangeli, Maria Laura De Angelis, Ann Zeuner

**Affiliations:** Department of Oncology and Molecular Medicine, Istituto Superiore di Sanità, 00161 Rome, Italy; marialaura.deangelis@iss.it

**Keywords:** colorectal cancer, intestinal stem cells, western-style diet, dietary factors, gut microbiota

## Abstract

Colorectal cancer (CRC) is the third commonly diagnosed cancer and the second leading cause of cancer-related deaths worldwide. Global CRC burden is expected to increase by 60% in the next decade, with low-income countries experiencing an escalation of CRC incidence and mortality in parallel to the adoption of western lifestyles. CRC incidence is also sharply increasing in individuals younger than 50 years, often presenting at advanced stages and with aggressive features. Both genetic and environmental factors have been recognized as major contributors for the development of CRC, the latter including diet-related conditions such as chronic inflammation and obesity. In particular, a diet rich in fat and sugars (Western-style diet, WSD) has been shown to induce multiple pathophysiological changes in the intestine linked to an increased risk of CRC. In this scenario, dietary factors have been recently shown to play novel unexpected roles in the regulation of intestinal stem cells (ISCs) and of the gut microbiota, which represent the two main biological systems responsible for intestinal homeostasis. Furthermore, diet is increasingly recognized to play a key role in the neoplastic transformation of ISCs and in the metabolic regulation of colorectal cancer stem cells. This review illustrates novel discoveries on the role of dietary components in regulating intestinal homeostasis and colorectal tumorigenesis. Particular focus is dedicated to new areas of research with potential clinical relevance including the effect of food components on ISCs and cancer stem cells (CSCs), the existence of CRC-specific microbial signatures and the alterations of intestinal homeostasis potentially involved in early-onset CRC. New insights on the role of dietary factors in intestinal regulation will provide new tools not only for the prevention and early diagnosis of CRC but also for improving the effectiveness of current CRC therapies.

## 1. Introduction

More than 2000 years ago Hippocrates, the father of modern medicine, stated that all the diseases begin in the gut. Recent studies support this theory by showing that alterations of intestinal homeostasis contribute to the development of chronic diseases, affect immune system function, and lead to cancer development [1,2,3,4].

Colorectal cancer (CRC) is tightly connected to alterations of the intestinal homeostasis and is closely linked to both individual lifestyle-related factors such as nutrition, smoking, alcohol consumption and physical activity and to social health determinants [5]. In recent years, the high rates of CRC incidence present in Western countries underwent a stabilization or slight decrease due to the spread of preventive screening practices in the population ≥50 years. However, at the same time colon cancer incidence in individuals younger than 50 years has been increasing both in Europe and in the USA [6,7], thus becoming a public health issue. Most cases of early-onset colorectal cancer (EO-CRC) are sporadic and are likely linked to environmental, behavioral and dietary factors, although large epidemiological studies are still lacking [8]. The gastrointestinal tract represents the primary site of nutrient absorption/digestion and the largest barrier to harmful toxins and pathogens between the host and its environment. Moreover, the intestinal tract is the largest endocrine organ of the body involved in the maintenance of metabolic homeostasis. Like other metabolically relevant organs, the intestine is susceptible to the effects of dietary factors. However, being in direct contact with food, the gut is primarily sensitive to dietary components, which directly impact both intestinal structure and function specifically affecting the stem cell compartment [9]. Basic intestinal units such as crypts and villi react dynamically to external inputs and have been shown to modify their size in response to dietary changes [10]. 

Recent studies demonstrated that dietary factors affect gut homeostasis at multiple levels, influencing both intestinal cells and the gut microbiota. In the mammalian intestine, unhealthy eating habits such as high fat diet (HFD), Western-style diet (WSD), intended as a diet rich in fat and simple carbohydrates, and overnutrition induce a plethora of changes ranging from alterations in enterocyte subcellular structures [11] to erosion of the whole crypt–villus organisation [12]. In particular, intestinal stem cells (ISCs) react to HFD or overnutrition with specialized adaptive responses that may be conserved throughout evolution. In fact, Drosophila ISCs respond to abundant food by activating a growth program based on accelerated division rates and predominance of symmetric divisions, resulting in a net increase of total intestinal cells and intestinal size [10]. Similarly, in the mammalian intestine overnutrition has been shown to directly boost ISCs proliferative activity through enhanced ß-catenin signalling, leading to increased villi length in the small intestine [13]. At the same time, HFD increases the level of bile acids that erode intestinal villi, leaving ISCs more exposed to toxic metabolites [12]. HFD-induced ISCs activation appears to be directly related to the lipid content of the food [14,15]. Recent studies showed that food with high lipid content induces robust Peroxisome Proliferator-Activated Receptor (PPAR-δ) activation in ISCs, thus enhancing the numbers of mitotically active stem cells in the intestine [15,16]. Importantly, besides eliciting transient adaptive responses in ISCs, HFD also increases the tumorigenicity of intestinal progenitors [9,15]. These observations are in line with several epidemiological studies showing that different diets are able to modulate the risk of intestinal cancer development [17,18]. Importantly, both mechanistic and population studies indicate that fat-rich diets such as WSD lead to a higher prevalence of CRC [19,20]. The intestinal microbiota is a crucial link between HFD and disease [21,22]. High-fat dietary supplementation has been shown to alter the abundance, composition, and physiological performance of the microbiota [23,24]. In turn, the gut microbiota has been shown to play an active part in fat storage and to crucially contribute to the pathophysiology of obesity [25,26]. This review summarizes a number of studies that contributed to shed light on the mechanisms linking dietary components to alterations of the main biological systems (cells and bacteria) responsible for intestinal health, eventually leading to altered homeostasis and CRC onset. Understanding the relationships between dietary factors, intestinal homeostasis and cancer is mandatory in order to develop successful strategies against the global CRC challenge. A deeper knowledge of diet-activated mechanisms in normal and neoplastic ISCs will be instrumental both to CRC prevention and to the management of millions of CRC patients awaiting more effective therapies and an improved quality of life.

## 2. Correlating Diet, Intestinal Homeostasis and Colorectal Cancer: Complexities and Limitations 

A number of studies highlighted the correlations between cancer risk and extrinsic factors such as specific dietary habits (non-Mediterranean Western diet, HFD, processed meat, alcohol consumption), sedentary lifestyle, environmental pollution and urbanization [27,28,29,30]. In particular, the adoption of WSD in both industrialized and developing countries is linked to higher obesity rates and prevalence of diabetes, cardiovascular disease and cancer, with a specific association between obesity and CRC [31]. 

The precise role of fat-rich diet in inducing CRC is still a matter of debate. In fact, mounting evidences emerging from cellular and animal studies point to a direct role of dietary fats in increasing colonic cell proliferation and tumorigenicity [12,13,15,16,32,33,34,35]. By contrast, a recently published meta-analysis reported a negative correlation between HFD and CRC risk [36]. This inconsistency can be partly attributed to the fact that molecular studies linking dietary patterns and cancer risk are relatively recent. Moreover, diet was not formally recognized as a cancer risk factor until 2015, when the 2015 Dietary Guideline Advisory Committee (DGAC) recognized an association (although considered moderate) between diet and CRC or post-menopausal breast cancer [37,38]. Most importantly, investigating the role of dietary patterns in human diseases is a challenging task due to the absence of universally standardized diets. In fact, diets that fall under a common definition often present substantial heterogeneity. For example, Mediterranean diet, WSD, and prudent dietary parameters vary between studies even though they use a similar nomenclature [39,40]. Concerning Mediterranean diet, it is particularly difficult to provide a univocal definition given the extent of the Mediterranean basin and the country-specific differences that characterize the eating patterns of resident populations.

A crucial issue in studies investigating the effects of dietary factors concerns the level of carbohydrates associated to a fat-rich diet, either in humans or in animal models [41]. In fact, the presence of carbohydrates within a HFD should always be specified, as the effect of lipids on intestinal homeostasis may be completely changed in the presence of sugars. An impressive demonstration of this statement comes from recent studies by Cheng and colleagues showing that a high-fat ketogenic diet promotes ISCs function and post-injury intestinal regeneration through beta-hydroxybutyrate (βOHB)-mediated Notch signalling. Strikingly, supplementation of HFD with glucose resulted in inhibition of 3-hydroxy-3-methylglutaryl-CoA synthase 2 (HMGCS2) (which produced beneficial ketone bodies) and reverted the favourable effects of ketogenic diet [42]. These findings reveal how diet-mediated control of βOHB signalling in ISCs helps to fine-tune stem cell adaptation during intestinal homeostasis and injury. Moreover, they provide further support to the hypothesis that the interactions between fats and carbohydrates crucially impact the regulation of intestinal homeostasis. In fact, several evidences suggest that a fat-rich diet (such as that followed by natives of extreme Northern regions) is not harmful *per se*, especially in the context of low environmental temperatures and/or elevated physical activity. Differently, the association of HFD with high levels of sugars (and additives) typical of Western diets may be principally responsible for alterations in intestinal homeostasis, particularly when linked to sedentary life, alcohol, smoke, pollution, and stress.

An evaluation of the specific role played by different types of fats in promoting CRC risk is also challenging. Polyunsaturated fats (n-3) are shown to be protective against colon cancer due to their anti-inflammatory properties [43]. By contrast, saturated fats have been shown to promote colon cancer risk [44]. On the other hand, saturated fats may also exert a protective role by increasing mucin production in some animal models of colon cancer [45]. The complexity of evaluating the disease-promoting effects of different diets and even of single dietary factors is further enhanced by the limitations of animal models used to reproduce human CRC. In fact, the majority of CRC genetic mouse models lead to the formation of adenomas that rarely progress to full carcinomas [46,47]. Second, adenoma formation in CRC genetic mouse models occurs in the small intestine, whereas human malignancies appear almost exclusively in the colon and rectum. Third, many conditions that promote CRC onset in humans, such as chronic inflammation [48] are not usually reproducible in mice. An important contribution in this field has been recently provided by Li et al., who explored the effects of WSD in rodents and observed a specific diet-induced reprogramming of leucine-rich repeat-containing G-protein coupled receptor 5 (LGR5) + ISCs and the subsequent occurrence of intestinal tumors [49]. Keeping in mind the complex challenges related to the investigation of dietary factors will help the interpretation of sometimes contrasting results in molecular and epidemiological studies linking diet and cancer.

## 3. Increased CRC Incidence in the Young Adult Population: Molecular Features and Potential Links with Dietary Habits

CRC has been traditionally considered a malignancy of aging adults. By contrast, an unprecedented increase of CRC cases in young adults is occurring in developed countries, often presenting at advanced stages and with aggressive features [50]. CRC incidence in individuals younger than 50 years is steeply increasing both in Europe and in the USA [6,7]. In Europe, CRC incidence increased by 7.9% per year among subjects aged 20–29 years from 2004 to 2016 while the increase in the age group of 30–39 years was 4.9% per year from 2005 to 2016 [7]. In the USA, CRC in young adults is expected to represent 10% of all colon and 22% of all rectal cancer cases, with a nearly doubled incidence of diagnoses in the population aged < 35 years [50]. Population studies confirmed similar trends also for Canada [51] and for Australia [52]. Among EO-CRC patients, 30% belong to families with genetic predisposition who require dedicated screening and management programs [28]. However, most cases of EO-CRC are sporadic and are likely linked to environmental, behavioral, and dietary factors [8]. EO-CRC is characterized by a more advanced stage at diagnosis, with a striking 76% of patients < 30 years diagnosed at a locally advanced or metastatic stage in contrast with 46–50% of older CRC patients [53,54,55]. Moreover, while in the last decade survival outcome has improved in late-onset CRC (LO-CRC), EO-CRC survival has not, its prognosis becoming even worse with decreasing patients age [56]. Despite its rising incidence and dismal prognosis, relatively few studies have addressed the molecular characterization of EO-CRC. The lack of information is particularly dramatic on colorectal tumors arising in very young patients (<35 years old), which appear to have peculiar molecular features and a different biological background [54,57]. Past studies pointed to some distinctive molecular features of EO-CRC as compared to LO-CRC. Such differences concentrate around genes that play key roles in diverse biological processes such as transcriptional regulation (FOS, FOSB and EIF4E), angiogenesis (CYR61) and protein catabolism (UCHL1) [58,59]. EO-CRC was also reported to be characterized by higher microsatellite instability and fewer B- rapidly accelerated fibrosarcoma (BRAF) V600 and neuroblastoma ras viral oncogene homolog (NRAS) mutations [60]. Latest wide genomic characterization studies reported that tumors from younger and older CRC patients have similar overall rates of genomic alteration but with significant gene-specific differences. In the microsatellite stable patient cohort, tumor protein 53 (TP53) and catenin beta 1 (CTNNB1) were more commonly altered in EO-CRC while adenomatous polyposis coli (APC), Kirsten rat sarcoma virus (KRAS), BRAF and family with sequence similarity 123B (FAM123B) were more commonly altered in LO-CRC. In the microsatellite instability high cohort, significant differences were seen in APC, BRAF and KRAS [61]. The mutational profile of KRAS, NRAS and Harvey rat sarcoma viral oncogene homolog (HRAS) were also recently analyzed in large patient cohorts, pointing to novel associations connecting age, gender and tumor subsite with specific mutations that could be potentially exploited for therapeutic purposes [62].

Recently, the description of four consensus molecular subtypes (CMSs) of colorectal cancer provided a means for grouping CRC cases with distinguishing features, with the goal of promoting more effective and personalized therapeutic strategies [63,64,65]. By applying CMS stratification to EO-CRC, Willauer and colleagues performed a retrospective study showing that patients younger than 40 years mainly fall within CMS1 (46%). CMS2 was relatively stable across age groups, while CMS3 and CMS4 were uncommon in EO-CRC patients (4% and 13%, respectively) [60]. Therefore, the majority of EO-CRC cases falls into the CMS1 and CMS2 subtypes. CMS2 tumors are characterized by epithelial differentiation and strong upregulation of wingless-related integration site (WNT) and BHLH transcription factor (MYC) downstream targets, classically implicated in intestinal epithelium organization and CRC carcinogenesis [63,66]. Differently, CMS1 (which encompasses the majority of microsatellite instability tumors (MSI) tumors) is characterized by an overexpression of proteins involved in DNA damage repair and, most importantly, by an increased expression of genes associated with a diffuse immune infiltrate along with strong activation of immune evasion pathways [67] Moreover, CMS1 is characterized by a polarized immune infiltrate mainly composed of type 1 T helper (TH1) and cytotoxic T cells that overturns adaptive immune responses, stimulates cancer cell proliferation and survival while promoting angiogenesis and metastasis [68]. The observation that the inflammatory CMS1 subtype is prevalent in young CRC patients underscores the importance of a comprehensive understanding of inflammation in the context of sporadic EO-CRC. Although more research is needed in order to reinforce the link between EO-CRC and dietary factors, several evidences suggest that Western diet and lifestyle play a key role in EO-CRC increase (Figure 1). First, EO-CRC incidence is increasing in parallel with overweight and obesity [51]. In a Swedish study, overweight during adolescence was associated with 208 times higher risk of developing CRC, and obesity with 238 times higher risk compared to normal weight [69]. Other diet components may play a role in EO-CRC incidence such as low vitamin A and vegetable intake [70], alcoholic drinks or processed meat [28]. Finally, it has been shown that EO-CRC incidence has raised across successive birth cohorts, suggesting the implication of exposures increasingly prevalent during childhood. Potential culprits are represented by cesarean delivery, absence of breastfeeding, prenatal/perinatal/infancy antibiotic use (which impacts the health of the gut microbiota) and changes in childhood diet [71]. Sedentary lifestyle during childhood, and specifically TV viewing time, has been also correlated with an increasing risk of EO-CRC [29] (Figure 1). In light of these findings, future strategies against EO-CRC must include a combined effort of epidemiological, basic and clinical research together with informative campaigns and socioeconomic measures. First, additional epidemiological studies are needed to clearly identify the risk factors specifically involved in EO-CRC development in order to allow targeted screening of specific population groups. Second, a widespread information strategy should be implemented in school children, adolescents and young adults in order not only to promote a healthy lifestyle and diet but also a specific awareness of EO-CRC and an early identification of symptoms (rectal bleeding, persistent change in bowel habits, abdominal pain, anemia). The dissemination of information on EO-CRC risks should ideally include pregnant women in pre-birth courses, in order to raise their awareness on the importance of natural delivery, breastfeeding and healthy baby diet on establishing early intestinal homeostasis. Finally, advancements in understanding the molecular determinants of EO-CRC are urgently needed, as well as multicenter clinical trials focused on young CRC patients and aimed both at optimizing currently available treatments and testing new therapeutic options.

## 4. The Intestinal Microbiota: A Key Player in Gut Homeostasis

The gut barrier is a functional unit organized as a multi-layer system formed by two main components (Figure 2): a physical barrier surface, which prevents bacterial adhesion and regulates paracellular diffusion to the host tissues, and a deep functional barrier able to discriminate between pathogens and commensal microorganisms and responsible for organizing the immune tolerance and response to pathogens. From the outer layer to the inner layer, the physical barrier is composed of gut microbiota (that competes with pathogens to gain space and energy resources, processes the molecules necessary to mucosal integrity, and modulates the immunological activity of the deep barrier), mucus (which separates the intraluminal content from more internal layers and contains antimicrobial products and secretory immunoglobulin A (IgA)), epithelial cells (which form a physical and immunological barrier), and the innate and adaptive immune cells forming the gut-associated lymphoid tissue (which is responsible for antigen sampling and immune responses). Disruption of the gut barrier has been associated not only with several gastrointestinal diseases, but also with extra-intestinal pathological conditions such as type 1 diabetes mellitus, allergic diseases or autism spectrum disorders [72]. The human body lives in symbiosis with a variegated microbial population comprising bacteria, archaea, viruses, fungi, yeasts and eukaryotic microbes, which altogether form the microbiota. This microbial population populates many human organs and tissues, including the gastrointestinal tract, the oral cavity and the skin [73]. While the human genome consists of approximately 23,000 genes, the microbiome encodes over three million genes, producing thousands of metabolites. The gastrointestinal tract contains as many bacteria as cells composing the whole human body [74] and intestinal bacterial products carry out crucial functions for the host’s fitness, phenotype and health [75,76]. Not surprisingly, in the last six years the number of scientific works focused on the microbiome has triplicated, revealing growing expectations on the microbiota as a potential preventive and therapeutic agent for multiple diseases.

The microbiota can be considered as a symbiotic living organism able to influence our body’s health. In line with this view, a state of generalized dysbiosis characterized by reduced microbial diversity and/or substantial shifts in resident species are associated to pathologic conditions ranging from neurologic disease to metabolic and cardiovascular disorders as well as gastrointestinal alterations and carcinogenesis [77,78,79,80,81,82,83]. 

The causes of intestinal dysbiosis are difficult to define due to their multifactorial nature. The variability of commensal microbiota can be influenced by age, sex, lifestyles, food habits, and prescription drugs. Considering its ability to influence the function of distal organs and systems, the gut microbiota resembles in many aspects an endocrine organ. In this role, the microbiota produces directly or indirectly numerous chemicals of hormonal nature that affect not only local entities such as the enteric nervous system, but are also released into the bloodstream and act at distal organs including the brain. For all these reasons the intestinal microbiota can be considered the larger and more biochemically heterogeneous endocrine organ in the human body [84].

In recent years, the gut microbiota has been the object of an increasing interest by the scientific community and has been associated with a large array of human diseases. Pathological conditions that have been linked (with variable strength) to microbiota alterations range from luminal diseases such as inflammatory bowel diseases (IBD) [85] and irritable bowel syndrome (IBS) [86] to metabolic diseases such obesity and diabetes [87], allergic diseases [88], neurodevelopmental disorders and cancer [79,82,83]. Host–microbe interactions and environmental factors (including diet) are recognized as major drivers in the co-evolution of the human–microbiome symbiosis [89]. Coevolution of the microbiome with its host is evidenced by the congruence of the phylogenetic trees of intestinal bacteria and primates [90]. Within the co-evolution process, the advent of Western lifestyle (involving urbanization, industrialization and WSD) represents a dramatic change, especially in light of its limited evolutionary timescale. 

Microbes living in the gastrointestinal tract of healthy individuals across the globe tend to adopt distinct community structures called enterotypes [91]. People living in different continents have different intestinal microbiota whose community structure is shaped by multiple factors including diet, environment, prescription drugs (such as antibiotics), and host genetics [92,93]. Although the gut microbiota tends to have community features related to the continent of origin, an interesting study showed that the microbiota of two Indian communities, one rural and one urban, were extremely different. The rural community had a microbiome characterized by a higher diversity as compared to the urban community and by a higher degree of intra-community homogeneity, highlighting the importance of the environment in shaping gut microbiota [94]. Moreover, microbiome studies performed in individuals from Botswana (which is a relatively wealthy country), Tanzania, and Philadelphia showed a greater similarity between the cohorts of Botswana and Philadelphia as compared to Tanzania, indicating that the gut microbiome is tuned with industrialization levels across populations [95]. Altogether, these observations provide evidence that the gut microbiota is a crucial determinant of intestinal health, contributing to the network that links dietary and environmental factors to human disease.

## 5. Metabolic Functions of the Gut Microbiota in Intestinal Development and Health

The gut microbiota, which derives its nutrients from host dietary components, is comparable to an organ endowed with extensive metabolic capability and substantial functional plasticity. The microbial population produces a series of metabolites performing numerous functions within the body. There is evidence that many different microbial metabolites also influence the host’s metabolism mostly by binding to specific cell membrane or nuclear receptors [96]. The gut microbiota provides essential capacities for the fermentation of non-digestible substrates like dietary fibres and endogenous intestinal mucus. Moreover, the microbiota produces a number of metabolites such as folate, indoles, secondary bile acids, trimethylamine-N-oxide (TMAO), but also neurotransmitters (e.g., serotonin, gamma aminobutiryc acid). Among these metabolites, the most abundant species are represented by short chain fatty acids (SCFA) such as acetate, butyrate and propionate [97,98] which represent both key signalling molecules and an important source of energy for the host [99]. Acetate, the most abundant SCFA, is metabolized by the liver [100] and reaches peripheral tissues, where it is involved in cholesterol metabolism and lipogenesis and may play a role in central appetite regulation [101]. Butyrate is also the main energy source for human colonocytes and has been shown to exert pleiotropic functions such as apoptosis induction of colon cancer cells, activation of intestinal gluconeogenesis and reinforcement of glucose and energy homeostasis [102,103]. Moreover, butyrate contributes to maintain oxygen balance in the gut by preventing microbiota dysbiosis [104]. Propionate is transferred to the liver, where it regulates gluconeogenesis and satiety signalling through interaction with the gut fatty acid receptors [103].

Epidemiological studies highlighted the relevance of the first 1000 days after conception (encompassing pregnancy and the first two years of life) for the healthy development of an adult. In this scenario, it appears that a correct establishment of the intestinal microbiota in the very first phases of life is essential for several vital functions, including maturation of the immune system. Labour and birth represent the first major exposure of the newborn to a complex microbiota. Moreover, due to the close vicinity of the birth canal and the rectum, natural delivery represents a primordial mechanism for intergenerational microbiota transfer in mammals, providing an efficient mechanism for the transmission of both vaginal and gut microbes. By contrast, rupture of the chorioamniotic membrane as occurs during cesarean section allows an exposure of the baby to a different set of maternal microbes (which are subsequently found in the meconium) resulting in an early alteration of the newborn microbiota [105]. Accordingly, it has been observed that cesarean delivery as well as intrapartum antibiotics during vaginal delivery alter bacterial colonization in neonates [106], although it is not completely understood which maternal strains colonize the different parts of the newborn’s body and their specific functions. Breastfeeding is the most relevant post-natal factor that supports adequate microbial colonization of the gut and drives immune system maturation [107,108,109]. Compared with formula feeding, breastfeeding has been associated with decreased morbidity and mortality in infants. Moreover, maternal milk and its microbiota are associated to a lower incidence of gastrointestinal infections and inflammatory, respiratory, and allergic diseases [109,110,111,112]. Breast milk is considered the gold standard nourishment for the infant. Beyond its undisputable role in newborns, breast milk also contains a wide variety of bioactive compounds that dynamically change their composition over time to satisfy the needs of the growing infant [113,114]. Milk also contains compounds such as urea and oxalate, two end-products of human metabolism that are indigestible for the baby but can be used as energy sources by beneficial microbes [115]. The types of glycans found in breast milk can shape the microbial composition of milk itself and of the infant gut microbiota, a phenomenon that has been investigated in detail for what concerns *Bifidobacterium* species [116]. After delivery, when the infant host defences are vulnerable, breast milk provides protection through the transfer of antimicrobial and anti-inflammatory compounds, while also stimulating the maturation of the immune system [107,108,109,110,112,117]. In addition, breast milk contains prebiotic compounds as well as its own microbiota, which cooperate to support the colonization and the microbial turnover in the infant gut.

The first period of life is also essential for a correct establishment of the interactions between the intestinal immune system and the microbiota. In fact, the immune system learns early to distinguish commensal bacteria (which are becoming almost-self and tolerated antigens) from pathogenic bacteria [118]. The gastrointestinal tract contains the largest and most active pool of immune cells present in the human body [119]. The interactions between intestinal immune cells and the gut microbiota play a reciprocal role in keeping intestinal homeostasis throughout life. In fact, commensal microorganisms are required for the proficient maturation of immune cells, while the immune system has a predominant effect on the composition of the microbiota [120]. The result of such reciprocal interaction is a perfect symbiosis between the human body and intestinal microorganisms. Moreover, interactions between gut microbes, intestinal cells and the immune system are increasingly regarded as regulators not only of energy metabolism but also of glucose and lipid homeostasis [121,122,123]. Dietary factors can dramatically influence microbioma health and diversity [124,125], reflecting on the function of the intestinal epithelium and immune system. Further understanding the links between diet, microbiota and the immune system will likely provide key directions for disease prevention and therapy. This is particularly true for CRC, which is one of the diseases most strictly correlated to an altered intestinal homeostasis.

## 6. Consequences of Altered Intestinal Homeostasis: Correlation between Dietary Factors, Microbiota and Intestinal Cancer

The mechanisms through which dysbiosis is proposed to affect tumorigenesis and/or tumor growth across cancer types are multiple and varied. Yet, comprehensive mechanistic insights are mostly lacking and studies are underway to understand how gut microbes may influence carcinogenesis in different organs. The largest body of evidence supporting a causal role for dysbiosis and intestinal inflammation as modulators of cancer development is available for CRC [126,127,128]. First, the tumor microbiota (including adenoma- and carcinoma-associated microbiota) is clearly different from that of the healthy intestinal mucosa [129,130,131]. Moreover, evidence from animal models shows that the transplant of stool from CRC patients can induce polyp formation, activate procarcinogenic signals and alter the local immune environment in mice as compared with stool derived from healthy controls [132]. 

Substantial changes in dietary habits have a dramatic impact on the gut microbiota, while the mechanisms responsible for diet-induced reduction of microbiota diversity are beginning to be elucidated. WSD has been shown to reduce gut microbiota diversity through multiple processes, including inflammation and the progressive loss of microbiota-accessible carbohydrates (MACs) found in dietary fiber. Importantly, changes associated to a diet low in MACs have been shown to occur over several generations. In fact, in mice consuming low levels of MACs, the reintroduction of dietary fiber is able to revert changes in gut microbiota in a single generation. However, upon a prolonged deprivation of MACs, microbiota alteration is not recoverable and remains stable over several generations [124,125]. HFD-induced inflammation also contributes to the decrease of bacterial diversity. HFD-related inflammation leads to the failure of adipocytes to effectively remove circulating free fatty acids and is pivotal to disease progression and the development of complications such as insulin resistance, cardiovascular disease, liver disease, atherosclerosis, intestinal disease, and cancer. Among the mechanisms linking diet, inflammation, and intestinal microbes, increased levels of *Firmicutes* and a reduced relative abundance of *Bacteroidetes* were observed in both humans and animals following HFD. In turn, shifts in gut microbiota populations activate Toll-like receptor (TLR) signaling pathway, leading to increased intestinal permeability to endotoxins and contributing to worsen the inflammatory state [133]. Increasing evidence from several preclinical models and from some human studies implicates dysbiosis as an oncogenic driver in CRC. In fact, dysbiosis leading to CRC development is characterized by the expansion of specific bacterial taxa such as *Fusobacterium nucleatum*, *Escherichia coli*, and *Bacteroides fragilis*. Some of the bacterial strains undergoing expansion during CRC have been shown to produce pro-inflammatory toxins promoting an inflammatory state that stimulates carcinogenesis [134,135]. *B. fragilis* produces enterotoxins associated to early-stage colorectal carcinogenesis in human subjects [136] and is specifically responsible for driving an inflammatory phenotype causing diarrhea and inflammation-related tumorigenesis [134,137]. Other bacterial metabolites with a directly genotoxic action, such as colibactin from *E. coli* or cytolethal distending toxin by *Campylobacter jejuni*, have been shown to induce carcinogenesis in mice [138]. Moreover, enterotoxigenic bacterial strains can contribute to intestinal inflammation by stimulating an increased production of reactive oxygen species and by affecting stress-related and DNA damage signaling pathways [139]. Among other strains, *F. nucleatum* has been demonstrated to play a central role in the development and progression of colon adenomas and colon cancer [140,141,142,143,144,145,146,147] and has also been detected in lymph nodes and distant metastases from CRC patients [148,149]. *F. nucleatum* has also been shown to potentiate gut tumorigenesis by inhibiting antitumor immune functions [140,150]. Moreover, *F. nucleatum* components including the FadA adhesion (FadAc) complex can activate the β-catenin–Wnt signalling pathway in human colon cancer cell lines, thus eliciting stem cell responses and resulting in oncogenic transcriptional changes [126,151]. A chronic inflammatory state may itself propagate dysbiosis, as demonstrated by the observation that genetic deficiencies in key inflammation-modulating genes promote the accumulation of protumorigenic bacteria, including *E. coli* [152,153,154,155].

Highly interesting correlations between microbiota alterations and cancer came from latest studies that altogether contribute to define CRC as a microbial disease. An important role in bacteria-mediated carcinogenesis is played by the secondary bile acid deoxycholic acid (DCA), which was previously known to increase colon cancer propensity [156]. An increase in the levels of DCA has been found in patients with low grade intestinal dysplasia, linking changes in the gut microbiome to the very early stages of CRC development [147,157,158]. DCA and other bile acids have been shown to increase DNA damage and promote the occurrence of mutations [159]. Also, increased levels of bile acids resulting from a fat-rich diet increase DNA damage and the proliferation of cancer stem cells (CSCs), contributing to the “bottom-up” model for CRC progression [12]. A recent meta analysis study by Wirbel and coworkers evaluated 768 fecal shotgun metagenomic studies of CRC, establishing disease-specific microbiome changes which are present across differences in environment, diet, and lifestyle [158]. Notably, this meta-analysis led to the identification of a set of 29 species indicative of CRC across populations from seven countries. Across different studies, the gut microbiome from CRC patients showed a functional enrichment in metabolic pathways involved in the degradation of amino acids, mucins, and organic acids. This finding is indicative of a metabolic shift towards aminoacid metabolism consequent to a fat- and meat-rich diet. By contrast, genes for carbohydrate metabolism were underexpressed in such conditions. Moreover, the authors found an elevated production of secondary bile acids from CRC metagenomes, suggesting a metabolic link between cancer-associated gut microbes and a fat- and meat-rich diet. In line with other studies showing gut microbiome signatures driven by specific diseases [160,161], Wirbel et al. established CRC-specific microbiome signatures distinct from other conditions such as type 2 diabetes, Parkinson’s disease, and inflammatory bowel disease. Notably, the gut microbiome of CRC cases showed a higher diversity as compared to controls, which was explained by the translocation of microbes from the oral cavity into the colon. Another seminal study by Maltez Thomas and colleagues showed an enrichment of gluconeogenesis, aminoacid putrefaction, and fermentation pathways associated with CRC. In contrast, metabolic pathways associated with complex carbohydrates, stachyose, and galactose were enriched in controls. The same study found a higher expression of genes related to trimethylamine (TMA) synthesis in CRC-associated metagenomes [157]. It is worth mentioning that the capacity of certain gut bacteria to degrade choline, found in meat and other foods, into TMA was previously involved in atherosclerosis [162]. The close relationship between gut microbiome and choline metabolism adds evidence to the mechanisms described by Wirbel et al. to confirm the role of potentially carcinogenic gut microbiota in CRC development. Altogether, the studies discussed in this section highlight the potential importance of gut microbial signatures for predicting CRC.

## 7. Effect of Dietary Factors on ISCs and on the Development of CRC

Diet is among the predominant environmental factors that affect human health, leading Hippocrates to state “Let food be your medicine and medicine be your food”. Not surprisingly, the gut is one of the most cancer-prone organs due to the high turnover of ISCs as well as their close contact with luminal mutagens, toxins, and harmful food-derived metabolites. Diet and its constituents play important roles in the regulation of inflammation [163], which in turn is implicated in the development of lifestyle-related chronic diseases such as cancer and cardiovascular diseases [164,165]. Dietary patterns can either promote or inhibit the onset of inflammation [166,167,168]. In fact, the Mediterranean diet and prudent dietary patterns generally produce anti-inflammatory effects [166,167,168]. Differently, high meat-intake dietary patterns such as WSD result in a pro-inflammatory effect [168]. Since food usually contains a mixture of both pro- and anti-inflammatory nutrients, a dietary inflammatory index (DII) has been developed in order to assess the potential effects of diet on an individual’s inflammation status [169,170]. The use of DII has improved the assessment of the role of dietary components in inflammation-related diseases and the risk of CRC onset [171,172]. The relationships between diet and intestinal inflammation have important implications for the regulation of ISCs. In fact, nutrient-sensing pathways and inflammatory pathways activated by dietary factors are involved in modulating the function of gut stem cells [9]. As an example, the low-grade inflammation and low levels of dietary vitamin D related to HFD have been shown to compromise the function of LGR5 + ISCs [173]. Furthermore, dietary factors have been shown to crucially influence the total number of ISCs and the mutational rate during stem cell divisions [174].

ISCs are not defined by an immutable phenotype but rather by a plastic cellular state that responds dynamically to perturbations. ISC plasticity includes the ability of non-stem intestinal cells to gain stem cell features, a process called de-differentiation [175,176,177,178,179]. Intestinal homeostasis is regulated at the stem cell level to maintain a tight balance between self-renewal and differentiation. Excessive self-renewal would expand the stem cell pool increasing the risk of tumorigenesis, but decreased self-renewal or inappropriate differentiation would shrink the ISCs pool mimicking a process of age-related degeneration that can also result in oncogenic transformation [180]. Although the molecular mechanisms driving cellular reprogramming remain to be fully elucidated, several studies have linked inflammation to loss of ISCs homeostasis and to the development of CRC. Seminal work by Schwitalla et al. demonstrated that inflammation-induced reprogramming could be responsible for the malignant transformation of intestinal cells. In fact, in a model of genetically induced inflammation based on constitutive Nuclear Factor-KB (NFKB) activation (resulting in activated Wnt and oncogenic KRAS signalling), postmitotic enterocytes were able to de-differentiate, acquire stem cell properties, and to initiate colon tumorigenesis [181]. Supporting the role played by nutrients in CRC development, a pioneer study by Newmark et al. reported that mice fed with a WSD (high in fat, low in vitamin D, calcium, and folate) developed intestinal tumors [19]. Recently, further insights linking nutrients, ISCs, and CRC were provided by Li and colleagues, who highlighted the involvement of nutrients in defining the contribution of two different stem cell populations (cycling LGR5 + ISCs and quiescent Bmi1 + ISCs) to mucosal homeostasis and tumorigenesis. The authors generated a mouse model of sporadic intestinal cancer consisting in feeding wild-type mice with a rodent WSD formulated to recapitulate nutrient intake linked to human WSD and higher CRC risk. Rodent WSD resulted in the appearance of intestinal tumors at an incidence and frequency similar to that in humans. With respect to the ISCs population, WSD reduced the number of LGR5 high ISCs and increased the Bmi + ISCs compartment as a compensation. Moreover, rodent WSD induced a complex transcriptional reprogramming of both stem cell populations including a mutational signature characteristic of replicative damage of human tumors, which was related to a lower intake of vitamin D3 and/or calcium [49]. Another interesting study highlighted the different role played by fat, proteins, and sugars in intestinal homeostasis. Ketone bodies (that can be produced from acetyl-CoA generated from fatty acid β-oxidation or from ketogenic amino acids) are produced at high levels in the intestine during periods of food deprivation and play an important role in the process of preserving and enhancing stem cell activity. A ketogenic carbohydrate-poor diet helps the intestine to maintain a large pool of adult stem cells, which are crucial for keeping the intestinal lining healthy. Notably, the authors found that, even in the absence of a fat-rich diet, ISCs can produce high levels of ketone bodies that activate the Notch signalling pathway and stem cell-related functions. By contrast, in the presence of a high-sugar diet, ketone production and stem cell function both declined [42]. Fatty acid components or their derivatives present in HFD enhance ISCs function also by modulating the levels of dietary cholesterol, which mediates phospholipid remodelling and tumorigenesis [32]. Other authors demonstrated that HFD can induce LGR5 expression, stem cell transformation, and colon carcinogenesis in a xenograft model of colon cancer independent of obesity through a vitamin A-bound serum retinol binding protein 4-stimulated by retinoic acid 6 (RBP4-STRA6) signalling pathway [35]. Moreover, intestinal progenitor cells exposed to HFD have been shown to acquire stem cell properties and to become more prone to oncogenic transformation through the activation of a PPAR-δ program [16]. As mentioned previously, a fat-rich diet increased the production of bile acids that are potent inducer of CRC [182]. The mechanism by which bile acids increase CRC risk involves the erosion of intestinal villi, leaving ISCs more exposed to toxic metabolites and increasing the possibility of malignant transformation. In the same study the authors found that animals with an APC mutation, the most common genetic mutation found in humans with CRC, developed cancer faster when fed with HFD. This event was mediated by the increased levels of bile acids resulting in the repression of farnesoid receptor X (FXR), a sensor of nutritional cues in ISCs [12]. Importantly these observations underline the convergence of both how dietary and genetic risk factors in determining CRC onset. 

Whereas the role of dietary factors in the development of CRC has received a wealth of attention in recent years, the role of diet in influencing CRC response to therapy, metastatization, and relapse is by far less understood. A previous study conducted in a large cohort of stage III CRC patients analyzed the influence of WSD (defined by high intake of red/processed meat, sweets and desserts, french fries, and refined grains) and prudent diet (high intake of fruits, vegetables, legumes, fish, poultry, and whole grains) on cancer recurrence and survival after postoperative adjuvant chemotherapy. Significantly, WSD was associated with a significantly worse disease-free survival [183], suggesting a direct influence of dietary factors on CRC progression. Adding to the evidence that diet can influence CRC patient survival, a recent observational study indicated that higher consumption of nuts (and specifically tree nuts) was associated with a significantly reduced incidence of cancer recurrence and death in patients with stage III colon cancer [184]. Understanding the effects of diet on CRC implies an in-depth analysis of the effect of dietary factors on colorectal CSCs, which are a cell population characterized by high plasticity and drug resistance [185,186]. Interestingly, the above-mentioned study by Karunanithi et al. investigated the effects of HFD on the CSCs compartment, showing that STRA6 or RBP4 downregulation in colon cancer cells specifically decreased the CSCs fraction and their sphere- and tumor-initiation frequency [35]. Moreover, supporting the beneficial effects of nuts on CRC, it was shown that walnut phenolic extract and its bioactive compounds, including (+)-catechin, chlorogenic acid, ellagic acid, and gallic acid downregulated the CSCs markers as well as β-catenin/p-GSK3β signaling and suppressed the self-renewal capacity of colorectal CSCs [187]. Recent studies also explored the role of HFD in CRC metastatization, showing that the activation of PPARδ (ligand dependent transcription factor involved in fatty acid metabolism) induces expansion of colonic CSCs and promotes CRC liver metastasis. Exposure to HFD induced expansions of the CSCs pool and promoted the expression of Nanog and CD44v6, which are related respectively to CSCs self-renewal and metastatic ability [188]. In particular, the PPARδ–Nanog axis was responsible for the effect of HFD in accelerating CRC liver metastasis [34]. Further studies are needed to define which dietary components are specifically responsible for activating proliferative and survival pathways in intestinal CSCs. Elucidating the role of CSCs and CSCs-related pathways would allow a more rational planning of dietary intake in CRC patients and would provide new tools to prevent CRC relapse.

## 8. Fasting, Caloric Restriction, and ISCs

Diet has a profound effect on tissue regeneration in diverse organisms, and low caloric states such as intermitting fasting have been shown to exert beneficial effects on age-associated loss of tissue functions. Accordingly, caloric restriction (CR) increases the lifespan and/or health in all investigated eukaryote species, including non human primates [189,190,191]. While HFD induces ISCs proliferation and transformation [16], CR and fasting apparently maintain the ISCs population activity without inducing tumorigenesis. Acute fasting regimens have pro-longevity and regenerative effects in diverse species, and they may represent a dietary approach to enhancing aged stem cell activity in multiple tissues [192,193,194]. Moreover, some forms of fasting may positively modulate both the microbiota and the immune system, acting either alone or in combination with drugs/biologicals against autoimmune diseases, neurodegeneration, and cancer [195]. Nutrient-sensing pathways are signalling systems involved in detecting intracellular and extracellular levels of sugars, amino acids, lipids, and surrogate metabolites and in transducing the presence of nutrients into molecular signals [196]. Nutrient-sensing pathways are commonly deregulated in human metabolic diseases and have been reported to play a key role in stem cells regulation and cancer. Main nutrient-sensing pathways relevant for tumorigenesis include the insulin/insulin-like growth factor 1 (IGF-1) signalling pathway, the target of rapamycin (mTOR) pathway, adenosine monophosphate-activated protein kinase (AMPK), and DNA-binding forkhead box O (FOXO) transcription factors [9,197]. After 6–24 h of fasting, a systemic response lowers the levels of glucose and insulin, increasing at the same time glucagon levels and the consequent production of ketone bodies. Fasting or fasting-mimicking diets (FMD) inhibit protein kinase A (PKA), thus increasing the activity of AMPK that activates early growth response protein 1 (EGR1), resulting in cell-protective effects. In cancer patients, it has been observed that fasting of at least 48 h has protective effects with respect to damage induced by chemotherapy in healthy tissues. Differently from fasting, which can be performed only for a limited time, CR can be protracted over time. On one side CR has been shown to exert beneficial effects on cardiovascular disease, diabetes, hypertension, and cancer, but on the other side, it can compromise overall health due to malnutrition-related effects [198]. The question concerning the use of fasting and CR in cancer therapy is a highly debated issue. Some authors are cautious concerning the use of short periods of fasting or CR in cancer patients, awaiting convincing clinical trial results. Moreover, due to cachexia, sarcopenia, and malnutrition often occurring in cancer patients, the use of such dietetic regimens is necessarily limited [199]. Other authors, while agreeing on the importance of performing controlled clinical trials to assess the validity of fasting, FMD, and CR in cancer, believe that such regimens should be explored and integrated with appropriate clinical evidence and drug-based therapeutic strategies due to their overall therapeutic potential [200]. Given the propensity of cancer cells, but not of normal tissues, to escape anti-growth signals due to oncogenic mutations [201] and their inability to adapt to fasting conditions [202,203], there is growing interest in the possibility that calorie-limited diets could become an integral part of cancer prevention and possibly of cancer treatment. Calorie-limited diets could also represent a means to increase the efficacy and tolerability of anticancer agents [202,203,204]. 

Targeting cancer metabolism is increasingly viewed as a potential strategy to eradicate cancer cells, leveraging the differences that characterize their altered metabolism as compared to that of normal cells. A recent study showed that lowering blood sugar levels through intermittent fasting in association with metformin inhibits the metabolic plasticity of cancer cells and has therapeutic effects in animal models. Intermittent fasting in combination with metformin triggers a chain reaction involving the activation of tumor suppressor PP2A (which act as an early sensor of energetic stress) that dephosphorylates and activates GSK3β, which in turn decreases the levels of the survival protein myeloid leukemia cell differentiation protein 1 (MCL-1), leading to cancer cell death [205]. Metabolic deregulation is recognized as a hallmark of cancer, and increasing evidences suggest that it can be exploited by neoplastic cells in order to resist harsh environmental conditions and to acquire a drug-resistant phenotype [206]. Metabolic plasticity enables cancer cells to switch their metabolism phenotypes between glycolysis and oxidative phosphorylation (OXPHOS) during tumorigenesis and metastasis, enabling cancer cells to tune their metabolic phenotypes in accordance to microenvironmental stimuli. As stem cells (both ISCs and CSCs) are characterized by high levels of plasticity [190], it is conceivable that they are also endowed with a significant ability to modulate their metabolism in order to optimize survival chances. A recent study provided evidence that fasting induces a metabolic switch in ISCs, substituting carbohydrate usage with fat burning. Interestingly, switching ISCs to fatty acid oxidation significantly enhanced their function. The metabolic switch occurred through the activation of PPARs, which activate the transcription of several genes involved in fatty acids metabolism. Inhibition of the PPAR pathway resulted in an inability of fasting to promote stem cell regeneration [207]. In addition to metabolic switching, a further aspect of metabolic plasticity is the fact that cancer cells can acquire a hybrid glycolysis/OXPHOS phenotype in which both glycolysis and OXPHOS can be used for energy production and biomass synthesis. The hybrid glycolysis/OXPHOS phenotype has been proposed to be specifically associated with metastasis and therapy resistance [208]. An important study by Yilmaz et al. reported that CR promotes ISC self-renewal indirectly through the inhibition of the mammalian target of rapamycin complex 1 (mTORC1) pathway in Paneth cells. Paneth cells in turn act as a regulatory niche to augment stem cell function in response to CR. Accordingly, Paneth cells isolated from calorie-restricted mice promoted the expansion of LGR5 + ISC in organoid culture more efficiently than cells from well-fed mice [33]. Nutritional status has also been reported to elicit differential responses of rapidly cycling ISCs and dormant stem cells to fasting. Phosphatase and tensin homolog (PTEN) and its inactive isoform phospho-PTEN are present in quiescent ISCs and are regulated by the phosphoinositide 3-kinases/protein kinase B (PI3K/AKT) pathway [209,210]. Changes in nutrient status leads to transient PTEN inhibition, rendering dormant ISCs functionally prompt to contribute to regeneration upon refeeding via activation of the PI3K/AKT/mTORC pathway [211,212].

Changes in circulating fatty acid levels induced by fasting also modulate ISCs function. Food withdrawal from mice increased ex vivo intestinal organoid formation, with LGR5 + ISCs being sufficient to drive this phenotype in a stem cell-autonomous and persistent manner. The same authors showed that fatty acid metabolism is an important modulator of ISCs function by identifying enriched targets of PPAR family members and fatty acid metabolism in RNA-sequencing of sorted ISCs from fasted and ad libitum fed mice. Among other proteins modulated by fasting in ISCs was carnitine palmitoyltransferase-1 (Cpt1a), a protein that catalyzes an essential step in long-chain fatty acid metabolism. Cpt1a induced by fasting prior to lethal irradiation protected ISCs and crypts in wild-type mice, while Cpt1a-inducible knockout mice have a decreased number of LGR5 + ISCs, lower proliferation of ISCs, and impaired organoid formation [207]. Inflammatory Bowel Disease (IBD) comprises various pathologies characterized by acute and chronic inflammation of the intestine that include rectal ulcerative colitis and Chron’s disease.

A recent study by Longo and colleagues demonstrated that FMD has significant effects in a dextran sodium sulfate (DSS)-induced murine model mimicking IBD. FMD alone could alleviate IBD symptoms, reducing systemic inflammation and leading to the decrease of protein C and leukocytes levels. FMD produced a substantial regeneration of the intestinal epithelium due to the activation of stem cells, to a greater control of the immune response, and to an increase of beneficial intestinal bacteria such as *Lactobacillaceae* and *Bifidobacteriaceae*. Notably, the results obtained with FMD were better than those obtained by fasting. Such results underline how specific nutrients included in the FMD eating plan (such as prebiotics) play a key role in counteracting inflammation and in regulating the intestine’s microbiota. Finally, the same authors provided evidence that only multiple cycles of FMD were able to promote ISCs activation and intestinal regeneration, suggesting that fasting primes the gut for improvement but that the refeeding phase of FMD triggers the regeneration of cells and tissues [213]. In summary, the metabolic properties of ISCs and CSCs represent a promising target for CRC prevention strategies and possibly for an improved treatment of CRC patients.

## 9. Conclusions

Dietary factors primarily influence intestinal homeostasis and CRC development, both directly through the action of food components on intestinal cells and indirectly by shaping the composition of the gut microbiota. The mechanisms linking dietary factors with neoplastic transformation are beginning to be elucidated and include multiple interactions between ISCs (and/or CSCs), the immune system, commensal bacteria and possibly other organs with metabolic/endocrine functions. The emerging mechanisms involved in intestinal homeostasis are opening new avenues of future research that may significantly improve the prevention and treatment of CRC: understanding the role of dietary and environmental factors in shaping intestinal health in prenatal/neonatal life and childhood; identifying gut microbial signatures that could be exploited for predicting CRC; deciphering the metabolic properties of ISCs and CSCs and their response to specific diets. Among these research fields, two seem particularly important given the increasing incidence of CRC and EO-CRC respectively in developing and industrialized countries. First, understanding the role of specific diets (including FMD) in influencing CRC response to therapy, metastatization and relapse could provide new tools to improve the treatment and management of CRC patients. Secondly, an improved understanding of the molecular determinants of EO-CRC and of diet-related risk factors are required in order to prevent the spreading of this deadly disease, together with new strategies of primary and secondary prevention and dedicated clinical trials. Altogether, forthcoming studies on how dietary factors orchestrate intestinal homeostasis are likely to provide new essential tools for an early diagnosis and a more effective treatment of CRC.

## Figures and Tables

**Figure 1 nutrients-11-02936-f001:**
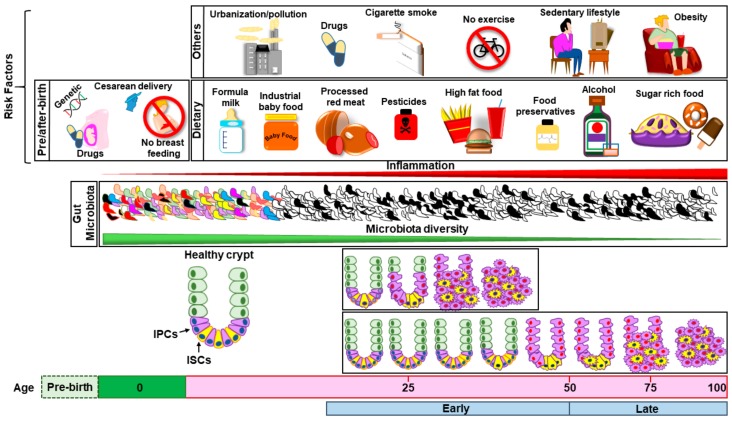
Effect of multiple risk factors on the development of colorectal cancer (CRC). Risk factors of different origin (environmental pollution, drugs, cigarette smoke, sedentary lifestyle) and diet-related risk factors are present immediately after birth and can influence the risk of early and late-onset CRC (upper part of the figure). Other factors acting in the prenatal/perinatal period including genetic predisposition, pre/perinatal antibiotics, cesarean delivery and absence of breastfeeding may also contribute to increase the risk of CRC development later in life. Different risk factors can increase the risk of CRC through multiple mechanisms (which are described in detail in the manuscript text) including enhancement of a pro-inflammatory state and decline of microbiota diversity (central part of the figure). Altogether, multiple factors act on the intestinal crypt (lower part of the figure) resulting in a transformation of intestinal stem cells (ISCs, yellow) and/or in a dedifferentiation of intestinal progenitor cells (IPCs, purple), gradually progressing into adenoma/carcinoma development.

**Figure 2 nutrients-11-02936-f002:**
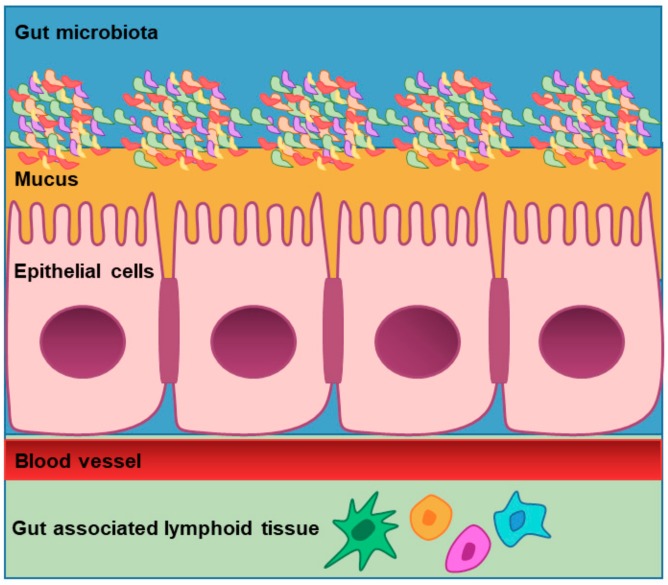
The gut barrier: From the outer layer to the inner layer, the intestinal barrier is composed of gut microbiota, mucus, epithelial cells, and the innate and adaptive immune cells forming the gut-associated lymphoid tissue.

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
