# Peer review of "Dietary Factors in the Control of Gut Homeostasis, Intestinal Stem Cells, and Colorectal Cancer"

_nutrients, 2019, doi:10.3390/nu11122936_

Round 1

Reviewer 1 Report

In this review, the authors summarize the current studies about the biological interactions between dietary factors (the western-diet, rich in fat and sugar), intestinal homeostasis and colorectal cancer (CRC). They provide evidences that the gut microbiota plays a key role in maintaining the intestinal health, that CRC is highly influenced by diet and that the higher increase in CRC incidence in young adults is potentially linked with dietary habits. Here, the fact of how a calories-limited diet can help the anti-cancer drug's efficiency is also addressed.

The review is particularly a good summary of the current understanding of the topic. The authors presented a large amount of detailed information and a broad selection of studies cited in the references. Each section of the review is properly discussed. Structure and flow are good. Importantly, the review outlines the best approaches for future research in colorectal cancer based, for example, on considering the gut microbial signature for predicting CRC, on a deeper study of the role of dietary factors in intestinal homeostasis in children and adolescents, and on studies that target cancer metabolism as a potential strategy to eradicate cancer cells and cancer stem cells.

The authors wrote in the Conclusions, line 659-661, “Unravelling the role of dietary factors in intestinal homeostasis will indicate more effective CRC preventive strategies particularly in children and adolescents, and will provide new tools for an improved management of CRC patients”. Perhaps, the addition of one or two sentences that recall what the authors wrote in section 6, 452-453, “potential importance of gut microbial signature for predicting CRC” ; section 7, 552-554, “Elucidating the role of CSCs and CSCs-related pathways would allow a more rational planning of dietary intake in CRC patients and provide new tools to prevent CRC relapse” ; section 8, 652-653, “In summary, the metabolic properties of ISCs and CSCs represent a promising target for CRC prevention strategies and possibly for an improved treatment of CRC patients”, makes the conclusions more detailed.

MINOR REVISIONS:

I would suggest a small extension of the Conclusions, in which I would include the future research avenues in the field. Avenues that have already been indicated in the individual sections of the manuscript.

My suggestion was to briefly recall the future research avenues, already discussed in the specific sections of the manuscript, in the Conclusions section.

Author Response

We thank the Reviewer for his positive evaluation of our work and for his constructive comments, which we have addressed in detail in the revised version of the manuscript.

We have extended the Conclusions and included a short description of future research avenues that link dietary factors, intestinal homeostasis and colorectal cancer.

Reviewer 2 Report

The current manuscript "Dietary factors in the control of gut homeostasis, intestinal stem cells and colorectal cancer (CRC)"  presents a detailed review of literature on contributing factors to CRC development and addresses the increasing incidence of early onset CRC that remains poorly understood.  It further examines the impact of dietary factors and their impact on dysbiosis in the gut microbiota that can lead to EO-CRC.  The review is well written and timely and will be relevant to those embarking on studies aimed at understanding the underlying factors that might be driving CRC and EO-CRC.

Reviewer 3 Report

The review article authored by Francescangeli et al., " Dietary factors in the control of gut homeostasis, intestinal stem cells and colorectal cancer." is a very well written. However, there are few sections that needs revision.

Comment:

1.) Abstract should be written in such a way that could highlight the important conclusion of the review.

2.) Conclusion should be more elaborate and based on references.

3.) There should be a subsection discussing about the future strategies to combat this deadly form of disease.

4.) Sub heading " Metabolic functions of the gut microbiota in intestinal development and health" is quite long and has diffused information. It needs to be more concise in order to consider broad range of readers.

Author Response

We thank the Reviewer for his positive evaluation of our work and for his constructive comments, which we have addressed in detail in the revised version of the manuscript.

The abstract has been modified in order to include some of the conclusive remarks of the manuscript. The Conclusions have been improved and extended, with reference to the different sections of the manuscript. We have added a subsection to the paragraph on early-onset colorectal cancer discussing potential strategies to counteract this deadly disease. The section “Metabolic functions of the gut microbiota in intestinal development and health” has been shortened and simplified for an improved comprehension by a broad audience of readers.